# Use of Digital Tools for the Assessment of Food Consumption in Brazil: A Scoping Review

**DOI:** 10.3390/nu16091399

**Published:** 2024-05-06

**Authors:** Adriane dos Santos da Silva, Flávia dos Santos Barbosa Brito, Debora Martins dos Santos, Amanda Rodrigues Amorim Adegboye

**Affiliations:** 1Nutrition Institute, Rio de Janeiro State University, São Francisco Xavier Street, 524, Rio de Janeiro 20550-900, Brazil; ane.adriane@hotmail.com (A.d.S.d.S.); deborams@gmail.com (D.M.d.S.); 2Centre for Agroecology, Water and Resilience (CAWR), Coventry University, Coventry CV8 3LG, UK; 3Centre for Healthcare Research, Coventry University, Coventry CV1 5FB, UK

**Keywords:** digital tools, survey, dietary intake, software, diet, Brazil

## Abstract

This is a scoping review on mapping the use of digital tools to assess food consumption in Brazil. Searches were carried out in nine electronic databases (Medline, Lilacs, Scopus, Embase, Web of Science, Science Direct, Ovid, Free Medical Journal and Crossref) to select studies published from October 2020 to December 2023. This review identified forty-eight digital tools in the 94 publications analyzed, the most frequent being web-based technologies (60%) and mobile devices (40%). Among these studies, 55% (*n* = 52) adopted a population-based approach, while 45% (*n* = 42) focused on specific regions. The predominant study design observed was cross-sectional (*n* = 63). A notable trend observed was the increasing frequency of validation studies in recent years. Although the use of digital tools in the assessment of food consumption in Brazil has grown in recent years, studies did not describe the process of creating and validating the tools, which would contribute to the improvement of data quality. Investments that allow the expansion of the use of the internet and mobile devices; the improvement of digital literacy; and the development of open-access tools, especially in the North and Northeast regions, are challenges that require a concerted effort towards providing equal opportunities, fostering encouragement, and delving deeper into the potential of digital tools within studies pertaining to food consumption in Brazil.

## 1. Introduction

Assessing food consumption of individuals or populations is a complex task [1]. It requires a methodology that facilitates data collection, ensures validity and improves the accuracy of the results [2,3].

The methods for assessing food consumption are, in general, based on the description and quantification of food consumed and data is recorded on paper. However, in recent years, technological advances in nutritional epidemiology have been driving the development and use of digital tools based on computers, mobile phones, cameras, scanners, personal digital assistants, and Web-based to replace paper-based instruments. These digital tools have improved automation and standardization of dietary assessment methods [4,5,6,7,8,9].

Studies have shown that digital tools have several advantages. For example, they allow for data collection optimization, greater storage capacity, information accuracy, speed in processing, and greater precision in estimating dietary data. They also reduce the cost and time spent on research, especially population-based research [2,5,7,8,10,11,12,13,14]. On the other hand, studies emphasize that the acceptability of digital tools for data collection may vary according to study population characteristics such as age, education, income, health conditions, and digital literacy [15,16,17,18,19].

Eldridge et al., in a literature review spanning from 2011 to 2017 and focusing on English-language publications, identified a total of forty-three digital tools designed for dietary assessment, with only two originating from Brazil [20]. Despite acknowledging various advantages associated with digital dietary assessment tools, the review highlighted a notable deficiency in the comprehensive documentation of the development, applicability, and validation processes of these tools, particularly within the context of food consumption assessment in Brazil [20]. Therefore, it is relevant to identify studies that developed, used, and tested digital technologies, based on web-based and mobile devices, in the assessment of food consumption, to describe in a systematic manner their characteristics, types of tools and their contribution to improve the assessment of dietary consumption in the country. Thus, the present study carried out a scoping review to map the use of digital tools for the assessment of food consumption in Brazil.

## 2. Materials and Methods

### 2.1. Study Design

This scoping review was carried out according to the review method proposed by the Joanna Briggs Institute (JBI) [21] and Arksey O’Malley, 2005 [22] and Levac Colquhoun and O’Brien, 2010 [23] and PRISMA-ScR (Preferred Reporting Items for Systematic Reviews and Meta-Analyses for Scoping Reviews) guidelines [24] [Appendix A]. The review process encompassed five sequential steps: formulation of the research question, development of the search strategy, study selection, data extraction, and synthesis and presentation of findings.

### 2.2. Eligibility Criteria

The inclusion criteria were formulated in alignment with the overarching research question. Adhering to the Population, Concept, and Context (PCC) strategy, the research question was structured as follows: “What is the purpose, extent, scope, and nature of digital tools utilized for assessing food consumption in the context of Brazil?” [21]. In this review we define digital tools as electronic platforms or applications specifically designed to collect, store, and analyse dietary intake data. These tools may include features like self-administered questionnaires, image capture, barcode scanning, and integration with dietary databases for nutrient estimation.

All types of publications available in full access, and which portrayed the topic of interest were eligible. Based on our definition of digital tool, studies solely relying on paper-based questionnaires that are later manually entered into spreadsheets or other software were not included. Studies that did not address aspects inherent to the research question, incomplete materials, studies not available in full, literature reviews and articles that were not in Portuguese, English or Spanish were excluded [22,23].

### 2.3. Search Strategy

An initial exploratory search was conducted to guide the development of the final search strategy. The comprehensive strategy was formulated following an examination of multiple article titles and abstracts deemed pertinent, as well as the indexing terminology employed in PUBMED. This strategy was then adapted for compatibility with other databases and tested before the definitive searches were executed.

The searches were performed by three independent reviewers (A.S.S.; D.M.S., F.S.B.B.), from October 2020 to December 2023, without time limit, in Medline (via Pubmed), Lilacs, Scopus, Embase, Web of Science, Science Direct, Ovid, Free Medical databases Journal and Crossref. In addition, these were searched on the CAPES journal portal, based on identification through the Federated Academic Community (CAfe), as a form of standardization for data collection. The PCC strategy, together with the terms Health Sciences Descriptors (DeCs), Medical Subject Headings (MESH), and Embase subject heading (Emtree) are described in Table 1. The full list of search terms is given in Appendix A.

Regarding the grey literature, the search for national dissertations and theses was carried out through the CAPES theses bank, Library of Theses and Dissertations (BDTD) coordinated by the Brazilian Institute of Information in Science and Technology (IBCIT), Google Scholar, the System for Information on Gray Literature in Europe (OpenGrey) and the Grey Literature network services (GreyNet).

The bibliographies of the papers included were examined manually, and forward citation searching was also conducted. Furthermore, contact was made with key researchers in the field of nutrition and identify additional studies.

### 2.4. Study Selection

All records were exported to the Zotero 6 reference manager to identify and quantify duplicates. After duplicate removal, the records were exported to Rayyan. Three investigators (A.S.S., D.M.S., F.S.B.B.) screened the titles and abstracts of the search results. Each abstract was independently reviewed by the investigators to assess eligibility according to the established inclusion criteria. Disagreements were resolved by group discussion and consensus. Full-text screening was performed independently and in duplicate by A.S.S., D.M.S., and F.S.B.B., and disagreements were resolved by consensus. The rationale for excluding a full-text study that did not meet the inclusion criteria was documented.

### 2.5. Data Extraction and Analysis

Data extracted included authorship, year of publication, type of study, geographic location, study population, type of tool and technology used. In addition, we extracted the information on the digital tool (time in minutes, free access) and validation process (reference method, main results), when available. Microsoft Excel version 16 was used for data collection and extraction. The extracted data were reviewed and summarized to meet the objectives of the review. Data from each paper were sorted into a summary and synthesized. Other characteristics of the included studies were summarized using graphs.

## 3. Results

The search strategy yielded 1287 articles. After eliminating duplicates, 1206 articles were screened based on their abstracts and titles. The full text of 424 articles was then reviewed and 94 articles were found to meet the inclusion criteria. Of these included studies, 56 (60%) were identified by an electronic search, 33 (35%) by searching the reference lists of the included studies, and 5 (5%) were recommended by consulted researchers. Figure 1 shows the PRISMA flow diagram for the screening process.

Of the 94 studies [8,9,25,26,27,28,29,30,31,32,33,34,35,36,37,38,39,40,41,42,43,44,45,46,47,48,49,50,51,52,53,54,55,56,57,58,59,60,61,62,63,64,65,66,67,68,69,70,71,72,73,74,75,76,77,78,79,80,81,82,83,84,85,86,87,88,89,90,91,92,93,94,95,96,97,98,99,100,101,102,103,104,105,106,107,108,109,110,111,112,113,114,115,116], 52 (55%) were population-based, and 42 (45%) were regional. Of these 42 studies, 26 (61.9%) were carried out in the Southeast region, as shown in Figure 2.

In total 44 (47%) papers were published in journals [8,9,25,26,28,29,30,31,50,51,52,53,54,55,57,59,63,65,69,79,82,83,85,86,87,88,89,91,92,93,96,97,98,102,103,104,105,106,108,109,111,112,113,116], 26 (28%) [34,35,36,37,38,39,40,41,42,43,44,45,46,47,48,49,71,72,73,74,75,76,77,78,100,101] were publications of population-based studies conducted by research institutes, 13 (14%) [27,32,33,60,61,67,81,84,90,95,99,110,115] were dissertations, 8 (8%) [58,64,66,68,70,94,107,114] were theses, and 3 (3%) [56,62,80] were research protocols. All included studies were published between 2004 and 2023. Regarding the type of study design, there was a predominance of cross-sectional studies and an increase in validation studies over the last six years (Figure 3).

The characteristics of all studies (*n* = 94) included in the scoping review are summarized in Appendix A. In terms of population, 58% of the digital tools were developed for adults, 35% for children and/or adolescents, and only 1% for the older adult population. In terms of the platform used, 60% were web-based, with 34% used free forms such as Google Forms and 40% were applications. For the data collection and entry interface, 35% of the studies used computers (notebooks and personal computers), 4% used personal digital assistants (PDAs), and 60% used mobile-based (smartphones and tablets) (Appendix A). Of all reported food consumption assessment methods, the 24-h recall (24HR) and food frequency questionnaires were the most common, with 33% and 31%, respectively (Figure 4).

The description of the 48 digital tools identified can be found in Appendix A. Of the 48, 58% the data entry method chosen was self-reported, in 25% the identification of food items was aided through digital images, and in 18%, the calculation of energy and nutrients was obtained through a database data integrated into the tool (Appendix A). Image capture as the method of assessing food consumption was used in only two studies [32,62]. The most frequent way of using images in research was to help identify household measures during the consumption report. The average time (minutes) to complete the dietary assessment was provided for 10 tools and it ranged from 3 min for Nova screener for the consumption of ultra-processed foods on the Epicollect5 Data Collection^®^ platform to 50 min for the Online questionnaire in personal digital assistant (PDA) (Appendix A).

Food composition data were reported as additional information in 63% of the tools. The most commonly used food composition tables in the studies were the Brazilian Food Composition Table (TACO) [117], the Nutritional Composition Table of Foods Consumed in Brazil published by the Brazilian Institute of Geography and Statistics [118], the new Brazilian Food Composition Table version 7.2 [119], and the United States Department of Agriculture (USDA) table, also known as the USDA Food and Nutrient Database for Dietary Studies (FNDDS) [120].

Of the 94 studies, 11 (11.7%) studies were related to tool validation. The characteristics of the validation of the tools included in the scoping review (*n* = 11) are summarised in Appendix A. In these studies, the most used reference or gold standard method was the 24HR and direct observation of food intake (Appendix A).

Several tools demonstrated satisfactory validity for assessing dietary intake, including the Online version of the Previous Day Food Questionnaire (PDFQ) for schoolchildren [63], Web-CAAFE for students [79], QUACEB for schoolchildren [96], and CUME online FFQ for adults [27]. The QSFA showed that the QSFA online was validates for iron and calcium [95]. The DQI-DFG tool exhibited consistent validity and reliability [50]. NutriSim was found to be user-friendly but with scattered answers [68] (Appendix A).

The FODMAP Project tool demonstrated good reproducibility for all FODMAP groups and good validity for lactose, with lower validity for other FODMAPs [115]. The CAAFE tool/questionnaire required further validation to improve its accuracy [59] (Appendix A).

## 4. Discussion

This scoping review on the use of digital tools to assess food consumption in Brazil identified 94 studies reporting 48 distinct digital tools. It was observed that that there is not a singularly preferred digital tool for food consumption assessment within the country. This is due to the complexity of data collection and the varying aims of included studies.

The most frequent dietary assessment methods were FFQ (35%) [25,27,28,54,63,66,71,72,73,74,81,88,92,95,105,112,113,114,115] and 24HR (31%) [9,29,31,51,55,56,58,61,68,78,80,85,89,96,107]. This finding corroborates the literature indicating these two methods are the most commonly employed approaches for evaluating food consumption patterns [2].

Due to the rapid advent of digital tools to assess food consumption in the international context, this scoping review was carried out to understand the Brazilian scenario regarding their development, purpose and applications.

Data from the 2019 Continuous National Household Sample Survey (Continuous PNAD) revealed that 82.7% of Brazilian households had Internet access, showing an increase of 3.6% compared to 2018. Additionally, approximately 96% of the Brazilian population utilized the Internet for communication purposes, such as sending or receiving text, voice or image messages [120]. Moreover, the percentage of individuals aged ten years or older who owned a personal mobile phone rose from 79.3% in 2018 to 81% in 2019 [120]. These figures underscore the growing use of the Internet and mobile phones across Brazil potentially indicating an emerging interest in digital technologies within the population.

The pandemic of severe acute respiratory syndrome (SARS) caused by the coronavirus (COVID-19), which affected all countries, accelerated the role of information and communication technologies (ICTs) in various aspects of our daily lives [121]. The increased use of such technologies has contributed to improving literacy and digital inclusion.

Given the current limitations of data collection by traditional methods, the increased use of technology can further boost the development and application of digital tools in studies that assess food consumption in the country. Research indicates that these digital solutions exhibit heightened accuracy, particularly among younger demographics equipped with computers or mobile devices, as well as among adults boasting higher levels of education and income [7,121]. Such findings highlight the transformative potential of technology in augmenting the efficacy and precision of dietary assessment methodologies, thereby underlining its pivotal role in advancing public health research and interventions.

The review found that studies were mainly conducted in the Southeast and South regions of Brazil. Santana et al. (2019) observed a rise in innovation funding from 2001 to 2014, particularly in the Southeast and South regions, while the North and Northeast regions received below-average funding.

Regarding internet access, the Continuous National Household Sample Survey—Continuous PNAD [120] indicates increased usage across all major regions, particularly in the Northeast, with a 5.2 percentage point rise. However, the Northeast still exhibits the lowest percentage of households with internet access (74.3%), potentially explaining the scarcity of studies utilizing digital tools in this region.

### 4.1. Validation Studies

The review observed a rise in validation studies conducted in Brazil in recent years. Of the eleven validation studies identified [27,50,59,63,68,79,90,95,96,107,115], only one underwent a validation process deemed unsatisfactory. While many studies reported satisfactory outcomes, some highlighted the need for further analyses across diverse age groups to enhance the tool’s accuracy.

Other studies have demonstrated the effectiveness of utilizing digital tools in assessing food consumption [14,20,122]. These tools not only help minimize errors but also ensure standardization in method application [14,20,122]. In Brazil, it was possible to identify that, in recent years, there have been promising advances in the development, validation, and application of digital tools to assess food consumption, for example, population-based studies such as The Consumer Expenditure Surveys (POF), The Study of Cardiovascular Risk in Adolescents (ERICA), the Brazilian National Survey on Child Nutrition (ENANI-2019) and, more recently, the cohort study NutriNet-Santé [9,29,78,113,123,124,125].

### 4.2. Comparison with International Studies

This review identified a promising trend in the development and use of digital food consumption assessment tools in Brazil. While the review by Eldridge et al. mapped 43 such tools globally, the present study identified 43 in Brazil alone, indicating a surge in domestic interest [20].

It is important to note that established reference methods, such as food weighing, biomarkers, and doubly labelled water, are still utilized in Brazil. However, the review found a relative scarcity of digital tools incorporating these gold-standard methods. Nevertheless, the observed increase in validation studies suggests potential improvement in this area, reflecting the commitment of Brazilian researchers to developing robust digital alternatives.

Examining the global landscape of digital food assessment tools reveals significant progress. The International Agency for Research on Cancer-World Health Organization (IARC-WHO) pioneered efforts with the development of EPIC-Soft, while ongoing advancements are evident in the FETA tool (FFQ EPIC Tool for Analysis) [126,127].

Brazil is one of the Latin American countries that are part of the EPIC-Soft initiative, which aims to adapt existing international methods for collecting individual food consumption data using the GloboDiet software, developed for use in different countries around the world [9]. The Brazilian version of the software, developed by Steluti et al., promises a more accurate assessment of national food consumption [9]. Another positive point is the information from the authors that the software will be made available, free of charge, to the scientific community on a platform for collecting short-term consumption data, through standardized and valid procedures. This availability could significantly contribute to the expansion of nutrition epidemiological studies carried out in the country, as only 2% of the identified tools in this review were freely accessible [9].

Due to the scarcity of validated and freely accessible digital tools, some studies used free digital forms such as Google Forms [25,33,84,88,89,96,105,109]. However, these forms present inherent limitations due to their structure and inability to comprehensively capture the complexities of food consumption assessment. These include difficulties in incorporating crucial elements such as food photographs, household measurements, and nutritional information about consumed items. There is also the impossibility of carrying out the instrument validation process.

Brazil is a country of continental dimensions, marked by social and economic inequalities and significant regional disparities. The results of this scoping review study can contribute to the description of the situation of digital tools in other developing countries with similar sociodemographic characteristics. Thus, compared to the international scenario, the process of development, use, and validation of new digital technologies remains limited.

### 4.3. Targeting Specific Populations and Assessment Methods

This review identified a focus on adult and adolescent populations in most studies, limiting applicability to children and older adults. However, initiatives are underway to address these gaps. Lacerda et al. developed a tool for assessing food consumption in children under five, contributing to updated national guidelines [82]. Similarly, Silva et al. are developing a tool for studies on older adult populations [111].

As Eldridge et al. highlight that selecting appropriate tools depends on study objectives and target audiences [20]. There is no single “gold standard” digital tool; researchers must consider each technique’s advantages, limitations, and applicability within a digital context [20]. Additionally, broader digital methods like food markers, dietary quality indicators, and food insecurity measures are valuable, particularly in large-scale studies like VIGITEL (Surveillance System of Risk and Protective Factors for Chronic Diseases by Telephone Survey for adults in Brazil) [34,35,36,37,38,39,40,41,42,43,44,45,46,47,48,49], PNS (National Health Survey) [75,76], Continuous PNAD [77], and PENSSAN network (National Survey on Food Insecurity in the Context of theCovid-19 Pandemic in Brazil) [100,101].

### 4.4. Limitations and Strenghths

This scoping review identified a significant increase in the use of digital tools for food consumption assessment in Brazil. However, several limitations are important to consider when interpreting the findings. This review focused on mapping the landscape of digital tools used in Brazilian dietary assessment studies. It did not assess the quality, validity, or effectiveness of the tools themselves. While the review identified a range of tools, it cannot comment on their specific functionalities, limitations, or potential for bias. The limited number of studies with detailed validation data restricts our ability to draw general conclusions about the overall validity of digital tools for dietary assessment in Brazil.

Few studies detailed described the methodological process of construction, the use of digital tools, or even provided details on whether the methods used to assess food consumption come from digital tools. The absence of this description in published papers reporting data on dietary assessment may have contributed to the underestimation of the number of studies included in this review and an underestimation of the actual number of digital tools currently in use within Brazil. Also, the review could not distinguish between tools originally developed as digital and those adapted from paper-based methods.

The development, implementation and maintenance of a digital tool require operational infrastructure and financial resources. Thus, the high cost and the need for technical skills to develop digital tools can also be identified as additional difficulties in the production of these tools in Brazil.

It is noteworthy that this review is one of the first to map the use of digital tools in studies on the assessment of food consumption in Brazil. Moreover, the list of references of included papers and grey literature databases contributed significantly to this review, especially the identification of relevant theses and dissertations. Furthermore, consultation with researchers in the field was an important step, representing 5% of the included studies. Despite the existing criticisms about the use of Google Scholar, 14% of the studies were identified via this database.

This review focused specifically on the Brazilian context. While findings may offer insights applicable to other developing countries with similar sociodemographic characteristics, further research is needed to explore the use of digital tools for food consumption assessment on a global scale.

### 4.5. Good Practice Guidelines

Based on good practice guidelines to assist and improve the quality of the description of new digital technologies in the assessment of food consumption proposed by Eldridge et al., it is important to emphasize that the recent national studies of digital tools to assess food consumption should include detailed information on the development, the purpose of the tool and applicability of the tool, type of technology used (computers, web-based, mobile devices, cameras, scanners, etc.), type of data entry (text, voice, image capture, barcode scanning, food list, food composition tables, etc.), chosen evaluation method (24HR, FFQ, etc.), financial resources and average time spent on development and in the data processing in their reports [20]. It is worth highlighting the importance of studies presenting, among their results, details on the usability, reproducibility, reliability, and validity of the tool being proposed.

## 5. Conclusions

This review identified a growing trend in the development and use of digital food consumption assessment tools in Brazil. However, several areas require further attention. Firstly, further studies are needed to enhance data availability and stimulate the development of new tools. Secondly, a national guide outlining best practices for tool selection would benefit researchers. Thirdly, it is recommended that the tools developed are properly validated, which is important to ensure a robust dataset, accurate results and the legitimacy of study findings. Finally, addressing the challenges of internet access, digital literacy, and technology development, particularly in underserved regions, is crucial for equitable application of these tools.

In conclusion, despite the identified limitations, the present review highlights the promising potential of digital food consumption assessment tools in Brazil. Continued domestic efforts in tool development and open-access initiatives, coupled with investments in infrastructure and digital literacy, hold significant promise for advancing nutritional research in the country.

## Figures and Tables

**Figure 1 nutrients-16-01399-f001:**
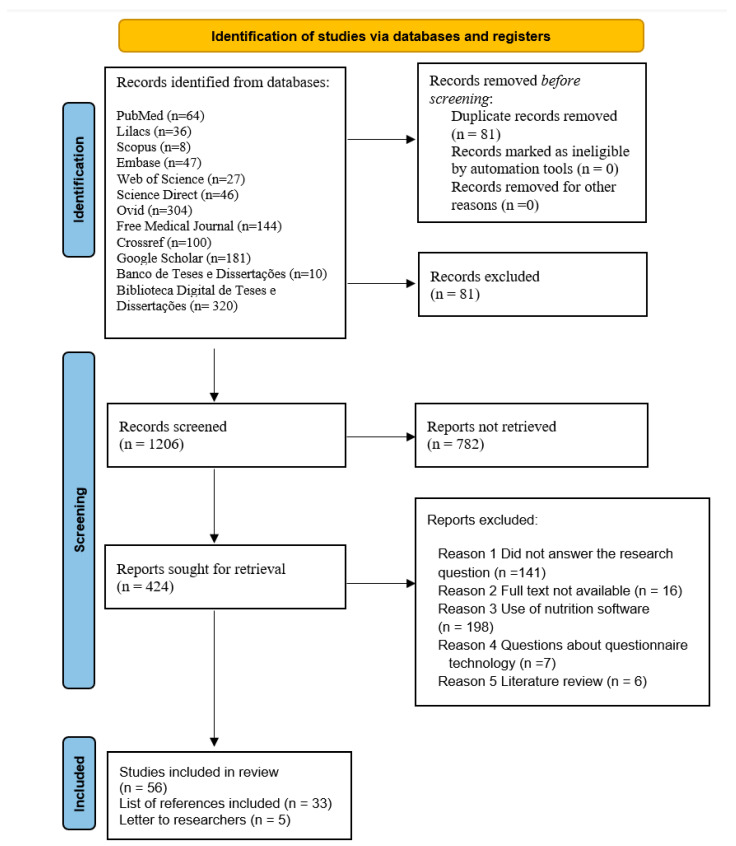
Process of identification and inclusion in this study: PRISMA flow diagram.

**Figure 2 nutrients-16-01399-f002:**
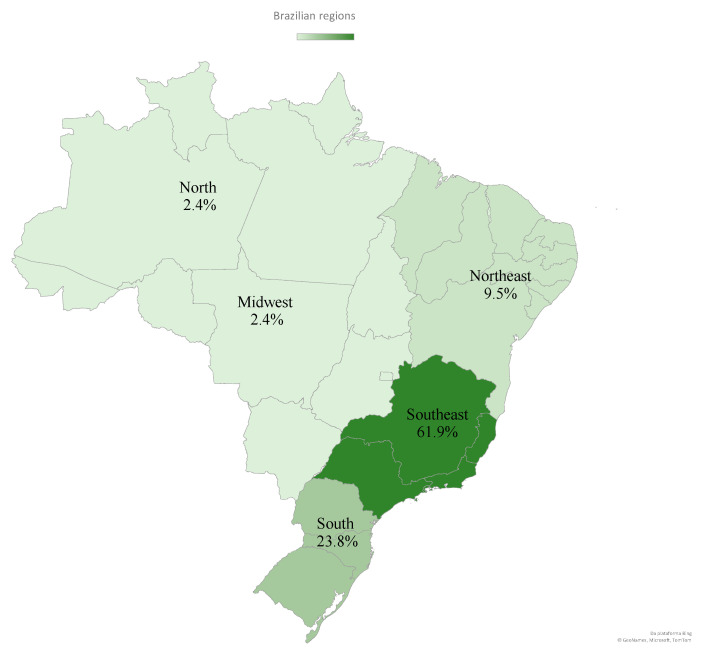
Distribution of the regional studies of the studies in Brazil by region.

**Figure 3 nutrients-16-01399-f003:**
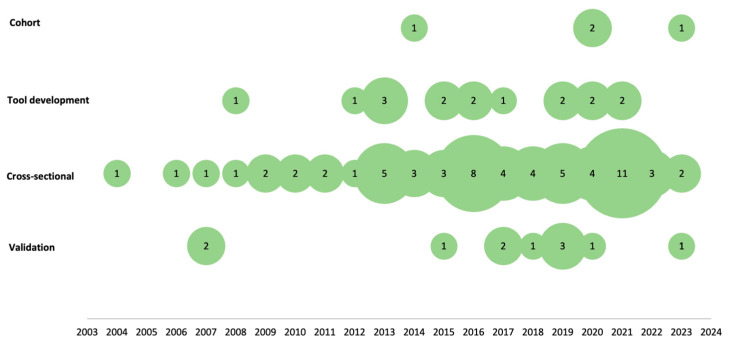
Distribution of the types of studies by year. (The numbers shown in the figure are the absolute frequencies (*n*) of the studies per year).

**Figure 4 nutrients-16-01399-f004:**
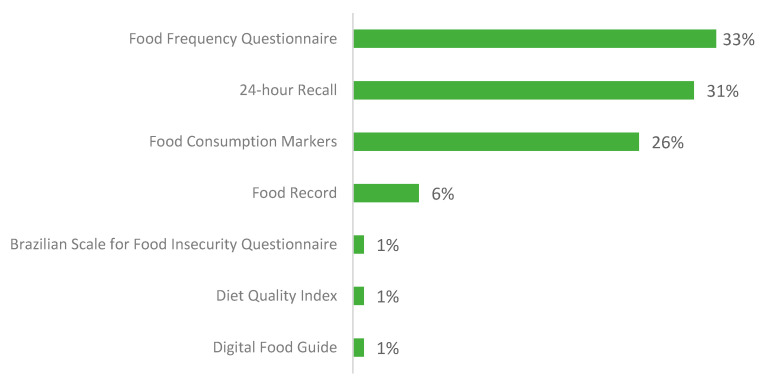
Distribution of the types of Food Consumption Assessment Methods.

**Table 1 nutrients-16-01399-t001:** Description of the PCC strategy and main terms used.

PCC Strategy	Decs Terms	MESH Terms	Emtree Terms
Population: Brazilian individuals of all age groups	adulto OR criança OR idoso OR adolescente OR gestante AND Brasil	adults OR child OR aged OR adolescent OR pregnant AND Brazil	adult OR child OR adolescent OR pregnant OR aged AND Brazilian
Concept: Digital questionnaires	“inquéritos e questionários” OR “inquéritos sobre dietas” OR inquéritos nutricionais AND software OR tecnologia	“Surveys and questionnaires” AND software OR electronic health records	technology OR software AND questionnaire
Context: Assessment of food consumption and nutritional surveys in Brazil	“Consumo de alimentos” OR “ingestão de alimentos” OR “registros de dieta” AND Brasil	“Diet records” OR “nutrition surveys” OR “nutrition assessment” OR “healthy surveys” OR eating AND Brazil	Brazil AND food intake OR dietary intake

Source: author. DeCs: Health Sciences Descriptors, MESH: Medical Subject Headings; Emtree: Embase subject heading.

## Data Availability

Not applicable.

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
