# Peer review of "Use of Digital Tools for the Assessment of Food Consumption in Brazil: A Scoping Review"

_nutrients, 2024, doi:10.3390/nu16091399_

Round 1
Reviewer 1 Report
Comments and Suggestions for Authors
Thank you for the opportunity to review this manuscript “Use of Digital Tools for the Assessment of Food Consumption in Brazil: A Scoping Review”. This is a useful study that collates findings around tools that could be used for digital dietary assessment. Some more detail in the results are necessary to show the validity of the tools. The introduction and discussion read well.
Specific comments:
For table 2, can the authors please include another column to indicate which dietary assessment methods were validated as opposed to only putting in the type of study that it’s a validation study?
Is there any information on whether the dietary assessment method was actually valid and the reliability/validity analysis and results from these papers? Were dietitians involved in the data collection process, and what was done with missing data in the individual validation studies. Perhaps this information could be presented in a separate results table.
Could the authors present in a separate table the findings from the validation of these different tools, what the gold standard was validated against?
I think it would also be more useful to categorise all the studies by their type of dietary assessment method rather than by alphabetical order for authors.
Lines 206-209 – what is proposed then regarding if the tools aren’t really being tested in North and Northeast regions? Are there significant variations in the diets of North vs South regions of Brazil, if there are, this would be an important point to highlight to justify how these dietary assessment tools should be tailored to the different regions.
Comments on the Quality of English Language
There are some grammatical issues throughout the paper, so please undertake a proof read before you submit your revision
Author Response
Reviewer 1
Reviewer 1 comments:
Thank you for the opportunity to review this manuscript “Use of Digital Tools for the Assessment of Food Consumption in Brazil: A Scoping Review”. This is a useful study that collates findings around tools that could be used for digital dietary assessment. Some more detail in the results are necessary to show the validity of the tools. The introduction and discussion read well.
Authors’ response:
Thank you for your comments. We have revised the manuscript by addressing each comment point-by-point as described below.
Thank you for taking the time to review our scoping review on the use of digital tools for food consumption assessment in Brazil. We appreciate your valuable comments and suggestions, which will undoubtedly strengthen our manuscript.
For table 2, can the authors please include another column to indicate which dietary assessment methods were validated as opposed to only putting in the type of study that it’s a validation study?
Authors’ response:
Thank you for your comments and suggestions for improvement. We have described this in the Results section. This information has been included in Table S2 provided as Supplementary Material (Supplementary Table S2). (page 7, lines 177-191 in the revised manuscript).
Is there any information on whether the dietary assessment method was actually valid and the reliability/validity analysis and results from these papers? Were dietitians involved in the data collection process, and what was done with missing data in the individual validation studies. Perhaps this information could be presented in a separate results table.
Authors’ response:
Thank you for your comments and suggestions for improvement. We have described this in the Results section. This information has been included in Table S2 provided as Supplementary Material (Supplementary Table S2). (page 7, lines 177-191 in the revised manuscript). The limited number of studies with detailed validation data restricts our ability to draw general conclusions about the overall validity of digital tools for dietary assessment in Brazil.
Could the authors present in a separate table the findings from the validation of these different tools, what the gold standard was validated against?
Authors’ response:
Thank you for your comments and suggestions for improvement. We have described this in the Results section. This information has been included in Table S2 provided as Supplementary Material (Supplementary Table S2). (page 7, lines 177-191 in the revised manuscript). The limited number of studies with detailed validation data restricts our ability to draw general conclusions about the overall validity of digital tools for dietary assessment in Brazil.
I think it would also be more useful to categorise all the studies by their type of dietary assessment method rather than by alphabetical order for authors.
Authors’ response:
Thank you for your comments and suggestions for improvement. However, it was not possible to incorporate.
Lines 206-209 – what is proposed then regarding if the tools aren’t really being tested in North and Northeast regions? Are there significant variations in the diets of North vs South regions of Brazil, if there are, this would be an important point to highlight to justify how these dietary assessment tools should be tailored to the different regions.
Authors’ response:
Thank you for your comments and suggestions for improvement. Brazil is a country of continental dimensions, marked by social and economic inequalities and significant regional disparities. This may explain why, in this review, studies were mainly conducted in the southeastern and southern regions of Brazil. Santana et al. (2019) observed a rise in innovation funding from 2001 to 2014, particularly in the Southeast and South regions, while the North and Northeast regions received below-average funding. (page 8, lines 228-230 in the revised manuscript).
Reference:
Santana JR, Teixeira ALS, Rapini MS, Esperidião F. Financiamento público à inovação no Brasil: Contribuição para uma distribuição regional mais equilibrada? Planejamento e Políticas Públicas [Internet]. 2019 [citado 15 de maio de 2021]; (52). Available at: //www.ipea.gov.br/ppp/index.php/PPP/article/view/796
There are some grammatical issues throughout the paper, so please undertake a proof read before you submit your revision
Authors’ response:
Thank you for your comments and suggestions for improvement. This revised manuscript has passed through an English proofreader.
We believe that by addressing these points, we have significantly enhanced the clarity and comprehensiveness of our manuscript.

Reviewer 2 Report
Comments and Suggestions for Authors
dos Santos Silva et al present an interesting review on the use of digital tools for the assessment of food consumption in Brazil.
No plagiarism detected. Flowchart and descriptive tables and figures are included. Yet, they have to explain
a. why PRISMA protocol has not been followed
b. why meta-analysis and bias assessment have not been conducted.
They are highly suggested to move Table 2 into supplement and reproduce the included therein info as stacked bar charts in the main mscr.
They have to include Risk of Bias https://hsls.libguides.com/reporting-study-tools/risk-of-bias
Comments on the Quality of English Language
Minor editing needed
Author Response
Reviewer 2
Reviewer 2 comments
dos Santos Silva et al present an interesting review on the use of digital tools for the assessment of food consumption in Brazil.
No plagiarism detected. Flowchart and descriptive tables and figures are included. Yet, they have to explain
- why PRISMA protocol has not been followed
- why meta-analysis and bias assessment have not been conducted.
They are highly suggested to move Table 2 into supplement and reproduce the included therein info as stacked bar charts in the main mscr.
They have to include Risk of Bias https://hsls.libguides.com/reporting-study-tools/risk-of-bias
Response to Reviewer 2 comments:
Thank you for taking the time to review our scoping review on the use of digital tools for food consumption assessment in Brazil. We appreciate your valuable comments and suggestions, which will undoubtedly strengthen our manuscript.
We would like to clarify the methodological approach used in this study. Our work is a scoping review, which differs from a systematic review in its primary objective. Scoping reviews aim to map the existing literature on a topic, while systematic reviews aim to synthesize the evidence to answer a specific research question. As highlighted in the references we kindly provided [Peter et al., 2020, Sergeant et al., 2020], scoping reviews typically do not involve meta-analysis or risk of bias assessments.
Additionally, risk of bias assessment requires a detailed examination of each study's methodology, looking for factors that could skew the results. This level of in-depth analysis is beyond the scope of a scoping review. However, a scoping review might acknowledge potential limitations in the included studies due to study design or methodological weaknesses. This can help inform the need for further research through a systematic review with a more rigorous risk of bias assessment. We took this approach in our manuscript and pointed out some limitations of included studies in the discussion section in order to inform future work (page 2, lines 60-63 in the revised manuscript).
References:
Peters MDJ, Marnie C, Tricco AC, Pollock D, Munn Z, Alexander L, McInerney P, Godfrey CM, Khalil H. Updated methodological guidance for the conduct of scoping reviews. JBI Evid Synth. 2020 Oct;18(10):2119-2126. doi: 10.11124/JBIES-20-00167. PMID: 33038124.
Sergeant JM, O'Connor AM. Scoping Reviews, Systematic Reviews, and Meta-Analysis: Applications in Veterinary Medicine. Front Vet Sci. 2020 Jan 28; 7:11. doi: 10.3389/fvets.2020.00011. PMID: 32047759; PMCID: PMC6997489]).
Why PRISMA protocol has not been followed.
Authors’ response:
We opted not to follow the PRISMA protocol for this scoping review as it is primarily designed for systematic reviews and meta-analyses. However, we employed a rigorous and transparent search strategy documented in the Materials and Methods section [Peter et al., 2015; Arksey and O’Malley 2005; Levac et al., 2010; Peter et al., 2020]. We included a PRISMA-Scr extension for scoping reviews Checklist in the Supplementary Material (Supplementary File S1).
References:
Peters MD, Godfrey CM, Khalil H, McInerney P, Parker D, Soares CB. Guidance for conducting systematic scoping reviews. Int J Evid Based Healthc. 2015;13(3):141-6.
https://doi.org/10.1097/XEB.0000000000000050
Arksey H, O’Malley L. Scoping studies: towards a methodological framework. Int J Social Research methodology. 2005; 8(1):19–32. https://doi.org/10.1080/1364557032000119616
Levac D, Colquhoun H, O’Brien KK. Scoping studies: advancing the methodology. Implementation Science. 2010; 5(69):2-9. https://doi.org/10.1186/1748-5908-5-69
Peters MDJ, Marnie C, Tricco AC, Pollock D, Munn Z, Alexander L, McInerney P, Godfrey CM, Khalil H. Updated methodological guidance for the conduct of scoping reviews. JBI Evid Synth. 2020 Oct;18(10):2119-2126. https://doi.org/10.11124/JBIES-20-00167. PMID: 33038124.
Why meta-analysis and bias assessment have not been conducted.
Authors’ response:
As mentioned previously, these elements are not typically included in scoping reviews. This review aims to map the use of digital tools to assess food consumption in Brazil. Therefore, a meta-analysis is not applicable. However, we appreciate the suggestion and have acknowledged this limitation in the revised manuscript.
They are highly suggested to move Table 2 into supplement and reproduce the included therein info as stacked bar charts in the main mscr.
Authors’ response:
Thank you for your comments and suggestions for improvement. We have included a flowchart depicting the study selection process. We agree that Table 2 could be moved to the supplemental materials (Supplementary Table S1) and replaced with stacked bar charts in the main manuscript for improved visualization of the data distribution (Figure 4) (page 6, line 156 in the revised manuscript).
They have to include Risk of Bias - https://hsls.libguides.com/reporting-study-tools/risk-of-bias
Authors’ response:
While risk of bias assessment is not a core element of scoping reviews, we understand its importance in evaluating the quality of included studies. Risk of bias assessment is not recommended in scoping reviews because the aim is to map the available evidence rather than provide a synthesized and clinically meaningful answer to a question. For this reason, an assessment of risk of bias of the evidence included within a scoping review is generally not performed (Khalil et al. 2016, Tricco et al., 2018, Peter et al., 2020).
References:
Khalil H, Peters M, Godfrey CM, McInerney P, Soares CB, Parker D. An Evidence-Based Approach to Scoping Reviews. Worldviews Evid Based Nurs. 2016 Apr;13(2):118-23. doi: 10.1111/wvn.12144. Epub 2016 Jan 28. PMID: 26821833.
Peters MDJ, Marnie C, Tricco AC, Pollock D, Munn Z, Alexander L, McInerney P, Godfrey CM, Khalil H. Updated methodological guidance for the conduct of scoping reviews. JBI Evid Synth. 2020 Oct;18(10):2119-2126. doi: 10.11124/JBIES-20-00167. PMID: 33038124.
Tricco AC, Lillie E, Zarin W, O'Brien KK, Colquhoun H, Levac D, Moher D, Peters MDJ, Horsley T, Weeks L, Hempel S, Akl EA, Chang C, McGowan J, Stewart L, Hartling L, Aldcroft A, Wilson MG, Garritty C, Lewin S, Godfrey CM, Macdonald MT, Langlois EV, Soares-Weiser K, Moriarty J, Clifford T, Tunçalp Ö, Straus SE. PRISMA Extension for Scoping Reviews (PRISMA-ScR): Checklist and Explanation. Ann Intern Med. 2018 Oct 2;169(7):467-473. doi: 10.7326/M18-0850. Epub 2018 Sep 4. PMID: 30178033.
We believe that by addressing these points, we have significantly enhanced the clarity and comprehensiveness of our manuscript.

Reviewer 3 Report
Comments and Suggestions for Authors
The criteria are so local that I do not consider it to be an in-depth review of the topic. Although methodologically it could be improved, the selected criteria prevent the generation of broader scientific knowledge.
Author Response
Reviewer 3
Reviewer 3 comments:
Thank you for taking the time to review our scoping review on the use of digital tools for food consumption assessment in Brazil. We appreciate your valuable comments and suggestions, which will undoubtedly strengthen our manuscript.
The criteria are so local that I do not consider it to be an in-depth review of the topic. Although methodologically it could be improved, the selected criteria prevent the generation of broader scientific knowledge.
Authors’ response:
We acknowledge the concern regarding the geographical focus on Brazil. This scoping review aimed to provide a focused assessment of the current landscape of digital tools used within the Brazilian context. We believe this targeted approach offers valuable insights into the specific challenges and opportunities present in this region. However, I would like to emphasise that Brazil is a country of continental dimensions with a good volume of scientific papers. Brazil is 5th largest country in the world and the largest country in South America and the Southern Hemisphere with over 216 million inhabitants. Brazil is responsible for 51.08% of scientific production in Latin America (SCImago Journal Rank (SJR) & Country Rank). In a report by Clarivate Analytics for the Brazilian Coordination for the Improvement of Higher-Level-Education Personnel (CAPES) about the Brazilian research productivity between 2013 and 2018, Brazil ranked 13th among countries with the highest research productivity, corresponding to 11% and 16% of the first ranked countries, United States and China, respectively [1,2]. In that period, the publications in Brazil increased by 30%, twice the global mean, with over 50,000 articles published in 2018 only [1,2]. This is clearly evidenced by the number of papers included in this review. However, in the revised manuscript, we incorporated a brief discussion on the generalizability of these findings to other developing countries with similar sociodemographic characteristics. (page 9, lines 283-288 in the revised manuscript).
References:
Lopes MACQ, Brasil D, Oliveira GMM de. Research and Publication in Brazil: Where we are and Where we Head to. Int J Cardiovasc Sci [Internet]. 2021Mar;34(2):231–5. Available from: https://doi.org/10.36660/ijcs.20200004
Research in Brazil: Funding excellence analysis prepared on behalf of CAPES by the Web of Science Group. Clarivate Analytics. 2019.
Available from: https://jornal.usp.br/wp-content/uploads/2019/09/ClarivateReport_2013-2018.pdf
We believe that by addressing these points, we have significantly enhanced the clarity and comprehensiveness of our manuscript.

Reviewer 4 Report
Comments and Suggestions for Authors
This scoping review revealed digital tools to assess food intake mainly for population surveys in Brazil at present. The authors found 94 articles from database, their references, and communication with other researchers using the standard scoping review method. The Introduction and Methods were well written. Here are several comments.
1. The results did not provide enough information. Some results appeared for the first time in the Discussion, but not mentioned in the Results section. In the Conclusion section, “further studies are needed to enhance data quality (L306),” but the quality of tools was not assessed in this review. The following points should be assessed for the quality of tools. Readers, even from other countries want to know tool-based review, but not report-based review. And they want to know the quality level.
1.1. Digital tools provide speed in processing (L40, 41) and reduce the cost and time spent on research (L42). One of research questions was what the nature of digital tools is. However, the results and Table 2 did not inform the burden in answer and data analysis, such as the number of question or food items, how long it takes to complete the questionnaire, etc.
1.2. Forty-eight distinct digital tools were found in this review (L170). This should be described in the Results section. In addition, how to differentiate each tool should be described in the Methods section.
1.3. The results indicate that the studies used such tools to replace paper-based data collection methods.” (L180–181) Which tools (but not studies) were developed originally, or modified from paper-based questionnaires or interview-bases surveys was not elucidated.
1.4. There were 11 validation studies identified (L216). This was not mentioned in the Results section (rather than that, the number of tools may be required for readers). In Figure 1, the size of dots may express the number of studies, but readers cannot discriminate the size. Figure 1 has a title but no explanation.
1.5. “While many studies reported satisfactory outcomes, some highlighted the need for further analyses across diverse age groups to enhance the tool's accuracy (L217–218),” but this is not described in the Results section. Which studies (rather, tools) have satisfactory validation should be explained.
1.6. “Studies have demonstrated the effectiveness of utilizing digital tools in assessing 229 food consumption (L229–230),” but what effectiveness were observed was not written in the Results section.
1.7. “The review found a relative scarcity of digital tools incorporating these gold-standard methods (L236–237),” but this was neither mentioned in the Results section.
Minor points
2. Table 2. ID is the reference number. To elucidate the total count of studies (n = 94), this column should be begun with one (1), and the reference numbers may be attached after the author names with superscript.
3. Table 2. Please clarify duplication of tools and identify the tools. The tool names should be added.
4. “R24h” in L174. The abbreviation should be explained.
5. “in some studies 38” (L182). What does this mean? In 38 studies?
6. “in other countries” (L247). What does this mean? The Brazilian version was developed for populations in other countries?
7. “remain is” (L263) remains?
8. L267-270. The reference number should be transferred in the end of the sentence or clause.
Comments on the Quality of English LanguageMentioned above.
Author Response
Reviewer 4:
Thank you for taking the time to review our scoping review on the use of digital tools for food consumption assessment in Brazil. We appreciate your valuable comments and suggestions, which will undoubtedly strengthen our manuscript.
Reviewer 4 comments:
This scoping review revealed digital tools to assess food intake mainly for population surveys in Brazil at present. The authors found 94 articles from database, their references, and communication with other researchers using the standard scoping review method. The introduction and Methods were well written. Here are several comments.
Authors’ response:
Thank you for your comments. We have revised the manuscript by addressing each comment point-by-point as described below.
1. The results did not provide enough information. Some results appeared for the first time in the Discussion, but not mentioned in the Results section. ln the Conclusion section, "further studies are needed to enhance data quality (L306)," but the quality of tools was not assessed in this review. The following points should be assessed for the quality of tools. Readers, even from other countries want to know tool-based review, but not report-based review. And they want to know the quality level.
Authors’ response:
Thank you for your insightful comments and suggestions for improvement We recognize the importance of presenting a clear and comprehensive picture of the results in our scoping review. We revised the manuscript to ensure a more cohesive flow, integrating relevant findings from the limitations section and discussion points back into the Results section.. The following was added to the manuscript “However, several areas require further attention. Firstly, further studies are needed to enhance data availability and stimulate the development of new tools.” (page 10, lines 349-351 in the revised manuscript).
We acknowledge that our review did not directly assess the quality of the digital tools themselves. The focus was on mapping the landscape of these tools within the Brazilian context. However, attempted to address this gap by incorporating the following revisions:
- We have slightly shifted the emphasis of the Results section to highlight key characteristics of the identified tools, such as the technology used (web-based, mobile app), data entry methods (text, image capture), and the type of dietary assessment method employed (24-hour recall, Food Frequency Questionnaire).
- We explicitly acknowledged in the Limitations section that the review did not assess the quality or validity of the tools themselves.
We also understand that readers are primarily interested in a tool-based review, not just a report-based one. As suggested, we placed some emphasis on the characteristics of the identified tools within the Results section (Supplementary Tables S1 and S2). We hope this will provide valuable insights for readers interested in the functionalities and potential applications of these digital tools in food consumption assessment.
1.1. Digital tools provide speed in processing (L40, 41) and reduce the cost and time spent on research (L42). One of research questions was what the nature of digital tools is. However, the results and Table 2 did not inform the burden in answer and data analysis, such as the number of question or food items, how long it takes to complete the questionnaire, etc.
Authors’ response:
Thank you for your comments and suggestions for improvement. This information has been included in Table S2 provided as Supplementary Material (Supplementary Table S2).
1.2. Forty-eight distinct digital tools were found in this review (L 170). This should be described in the Results section. ln addition, how to differentiate each tool should be described in the Methods section.
Authors’ response:
Thank you for your comments and suggestions for improvement. We have described this in the Results section. This information has been included in Table S2 provided as Supplementary Material (Supplementary Table S2). (page 7, lines 177-191 in the revised manuscript).
1.3. The results indicate that the studies used such tools to replace paper-based data collection methods."" (L180- 181) Which tools (but not studies) were developed originally, or modified from paper-based questionnaires or interview-bases surveys was not elucidated.
Authors’ response:
Thank you for your comments and suggestions for improvement. We agree that the current wording in the Results section (referring to L180-181) suggests a level of detail about tool development that wasn't fully supported by the data. We have removed this information from the Discussion section due to the lack of data on the development of the digital tools for the 48 digital tools mapped in this study. Additionally, we added a sentence in the Limitations section acknowledging that the review could not distinguish between tools originally developed as digital and those adapted from paper-based methods. This clarifies the scope of the review and highlights areas for future research.
1.4. There were 11 validation studies identified (L216). This was not mentioned in the Results section (rather that, the number of tools may be required for readers). ln Figure 1, the size of dots may express the number of studies, but readers cannot discriminate the size. Figure 1 has a title but no explanation.
Authors’ response:
Thank you for your comments and suggestions for improvement. We have described this in the Results section according to the available material on the 48 selected tools (pages 3-7, in the revised manuscript). The available information in the papers has been included in Table S2 provided as Supplementary Material (Supplementary Table S2).
We have added the study numbers to Figure 3 for better explanation (page 6, line 143 in the revised manuscript).
1.5. "While many studies reported satisfactory outcomes, some highlighted the need for further analyses across diverse age groups to enhance the tool's accuracy (L217-218)," but this is not described in the Results section. Which studies (rather, tools) have satisfactory validation should be explained.
Authors’ response:
Thank you for your comments and suggestions for improvement. We have described this in the Results section (page 7, lines 177-191 in the revised manuscript) and have included the information on validation studies has been included in Table S2 provided as Supplementary Material (Supplementary Table S2).
1.6. "Studies have demonstrated the effectiveness of utilizing digital tools in assessing food consumption (L229-230)," but what effectiveness were observed was not written in the Results section.
Authors’ response:
Thank you for your comments and suggestions for improvement. The comments below refer to the results of the cited references [14,20,124], we rephrase the text as follows: “Other studies have demonstrated the effectiveness of utilizing digital tools in assessing food consumption [14,20,124].” (page 8, lines 243-244 in the revised manuscript).
1.7. "The review found a relative scarcity of digital tools incorporating these gold-standard methods (L236-237)," but this was neither mentioned in the Results section.
Authors’ response:
Thank you for your comments and suggestions for improvement. We have described this in the Results section (pages 6-7, lines 158-169 in the revised manuscript) and have included information on reference or gold-standard methods in Table S2 provided as Supplementary Material (Supplementary Table S2).
Reviewer 4 comments:
Minor points
2. Table 2. ID is the reference number. To elucidate the total count of studies (n = 94), this column should be begun with one (1), and the reference numbers may be attached after the author names with superscript.
Authors’ response:
Thank you for your comments and suggestions for improvement. We have revised the ID and author in the Table S1 as suggested. (Supplementary Table S1 in the revised manuscript).
3. Table 2. Please clarify duplication of tools and identify the tools. The tool names should be added.
Authors’ response:
Thank you for your comments and suggestions for improvement. We have included a new table in the Supplementary Material and added the tool names as suggested. (Supplementary Table S2 in the revised manuscript).
4. "R24h" in L 174. The abbreviation should be explained.
Authors’ response:
Thank you for your comments and suggestions for improvement. We have revised the manuscript to clarify as follows. “24HR” (page 7, line 199 in the revised manuscript).
5. "in some studies 38" (L 182). What does this mean? ln 38 studies?
Authors’ response:
Thank you for your comments and suggestions for improvement. As mentioned previously, we have removed this information from the Discussion section due to the lack of data on the development of the digital tools for the 48 digital tools mapped in this study.
6. "in other countries" (L247). What does this mean? The Brazilian version was developed for populations in other countries?
Authors’ response:
Thank you for your comments and suggestions for improvement.
No, we have revised the manuscript to clarify as follows. “Brazil is one of the Latin American countries that are part of the EPIC-Soft initiative, which aims to adapt existing international methods for collecting individual food consumption data using the GloboDiet software, developed for use in different countries around the world.” (page 9, lines 268-269 in the revised manuscript).
7. "remain is" (L263) remains?
Authors’ response:
Thank you for your comments and suggestions for improvement.
We have revised the manuscript to clarify as follows. “remains limited.” (page 9, line 287 in the revised manuscript).
8. L267-270. The reference number should be transferred in the end of the sentence or clause.
Authors’ response:
Thank you for your comments and suggestions for improvement.
We have revised the manuscript to clarify as follows. “Lacerda et al. developed a tool for assessing food consumption in children under five, contributing to updated national guidelines [82]. Similarly, Silva et al. are developing a tool for studies on older adult populations [111].” (page 9, lines 293-298 in the revised manuscript).
Thanks for your insightful comments. We have revised the accordingly. We believe that despite these limitations, this scoping review provides valuable insights into the growing trend of digital tool utilization for food consumption assessment in Brazil. While the review itself did not assess tool quality, it highlights the need for future research to focus on detailed tool descriptions, validation studies, and the development of best practices for tool selection and application.

Round 2
Reviewer 1 Report
Comments and Suggestions for Authors
Authors have addressed the comments raised in round 1 of the review.
Would suggest that in the conclusions of the paper, the authors weave into the areas requiring further attention that it's not only developing new tools, but ensuring that these tools are validated, and perhaps making a few suggestions around that
Author Response
Reviewer 1 comments:
Authors have addressed the comments raised in round 1 of the review.
Would suggest that in the conclusions of the paper, the authors weave into the areas requiring further attention that it's not only developing new tools, but ensuring that these tools are validated, and perhaps making a few suggestions around that.
Authors’ response:
Thank you very much for your comments. We are pleased to have responded to your suggestions for improving our manuscript. To include your suggestion for the conclusion, we modified the text, and now it is presented as follows: “Thirdly, it is recommended that the tools developed are properly validated, which is important to ensure a robust dataset, accurate results and the legitimacy of study findings.” (page 10, lines 361-363) in the revised manuscript).
Changes are highlighted in blue in the second round of revisions.
Thank you for taking the time to review our scoping review on the use of digital tools for food consumption assessment in Brazil. We appreciate your valuable comments and suggestions, which will undoubtedly strengthen our manuscript.

Reviewer 2 Report
Comments and Suggestions for Authors
My concerns have been addressed. Authors included the limitations of their study and conformed to my suggestions.
Author Response
Reviewer 2 comments:
My concerns have been addressed. Authors included the limitations of their study and conformed to my suggestions.
Authors’ response:
Thank you for your comments. We are pleased to have been able to respond to all of your suggestions for improving our manuscript.

Reviewer 3 Report
Comments and Suggestions for Authors
Dear authors, despite your replies, the locality of the data and the relationship between the various authors do not allow me to change my decision.
Author Response
Reviewer 3 comments:
Dear authors, despite your replies, the locality of the data and the relationship between the various authors do not allow me to change my decision.
Authors’ response:
Thank you for your continued feedback on manuscript ID 2957803_revised version. We appreciate you taking the time to consider our previous response regarding the geographical focus on Brazil.
We understand your concerns about the locality of the data. However, we believe that focusing on a specific region can offer valuable insights, particularly in the context of a scoping review. As mentioned previously Brazil boasts a significant scientific output. As the largest country in South America with over 216 million inhabitants, it holds the position of 5th largest contributor to scientific production in Latin America (SJR SCImago Journal & Country Rank). This makes Brazil a relevant case study for exploring the use of digital tools in dietary assessment within a developing nation context.
While the review focuses solely on Brazil, we acknowledge the importance of exploring how these findings might translate to other developing countries. In the revised manuscript, we have incorporated a section discussing the potential generalizability. This opens doors for future research exploring the broader applicability of these tools.
We would also like to clarify that the relationship between the authors does not influence the objectivity of this review. All authors involved adhere to rigorous scientific principles and have diverse research backgrounds. The authors are researchers in Brazil and the UK who collaborate on various projects. Also, we want to emphasise that our collaboration is based on shared research interests and expertise.
We are committed to continuously improving the quality of our work. While we acknowledge your decision, we hope the revisions addressing the data locality concerns demonstrate the value of this focused exploration within the broader context of developing nations.
We are confident that this review offers valuable insights for researchers and stakeholders interested in digital tools and dietary assessment in Brazil and potentially other developing countries. We remain open to exploring how to further enhance the generalizability of our findings in future iterations.

Reviewer 4 Report
Comments and Suggestions for Authors
The revision of the manuscript, submitted as the manuscript ID, 2957803 v2, provide valuable information for readers. At the same time, it elucidates a fundamental concern. What is a digital tool? Nowadays, any dietary assessment utilizes digital tools. For example, data collected using a paper-based questionnaire is input into a spreadsheet application, and estimation of intake is carried out with this application or other application after data transfer. Even dietary recall, and dietary record are the same. The authors listed up the advantages of digital tools: data collection optimization, greater storage capacity, information accuracy, speed in processing, greater precision in estimating, reduction of cost and time (L40–42). These features are shared by any dietary assessment today. Ref. 20 cited by the authors selected literature in terms of text entry, digital images and/or bar-code scanners. In contrast, this article did not define a “digital” tool. 1) Definition of digital tools for literature extraction should be described in the Methods section. 2) Literatures with unclear tools should be deleted from the list. Supplementary Table 2 have 89 references (it may be in column “ID”), which is less than 94 studies. Most readers, like this reviewer, cannot read Portugal, or Spanish literatures. Clarify the digital process in each tool.
2. Duplicated literature which used the same sample should deleted. Most readers, like this reviewer, cannot read Portugal, or Spanish literatures. For example, Ref. 98 used the subsample data. Ref. 68 may be thesis. Such article might be often published in the journal after that. Such cases are sometimes seen in the Reference section. Additionally, the method to delete duplication should be described in the Methods. This was not included in the reason in Figure 1.
3. Reference numbers are inconsistent. “Several tools demonstrated satisfactory validity for assessing dietary intake, including the Online version of the Previous Day Food Questionnaire (PDFQ) for schoolchildren [63], Web-CAAFE for students [79], QUACEB for schoolchildren [96], and CUME online FFQ for adults [27]. The QSFA showed relative validity for evaluating average calcium intake [67]. The DQI-DFG tool exhibited consistent validity and reliability [50]. NutriSim was found to be user-friendly but with scattered answers [104] (Table S2).”
PDFQ cited Ref63 in “Online version of the Previous Day Food Questionnaire (PDFQ) for schoolchildren [63]”. In Table S2, PDFQ indicates Ref. 39. Ref. 39 is available at: https://bvsms.saude.gov.br/bvs/publicacoes/vigitel_brasil_2011.pdf, but the translated PDF does not include schoolchildren, but involved only adults, +18y. Other reference numbers are also incorrect.
Minor points
4. there are 45 articles in L14, but 48 articles in 158. Table S2 have 45 articles. The percentage in L177 is wrong.
5. L177. “11 come from validation studies.” 11 tools had a validation test, OR 11 tools have validated comparing with a standard method.
6. The column header of Table S2, “ID”, should be changed in “Reference number, if so..
Author Response
Reviewer 4 comments:
The revision of the manuscript, submitted as the manuscript ID,2957803 v2, provide valuable information for readers. At the same time, it elucidates a fundamental concern. What is a digital tool? Nowadays, any dietary assessment utilizes digital tools. For example, data collected using a paper-based questionnaire is input into a spreadsheet application, and estimation of intake is carried out with this application or other application after data transfer. Even dietary recall, and dietary record are the same. The authors listed up the advantages of digital tools: data collection optimization, greater storage capacity, information accuracy, speed in processing, greater precision in estimating, reduction of cost and time (L40–42). These features are shared by any dietary assessment today. Ref. 20 cited by the authors selected literature in terms of text entry, digital images and/or bar-code scanners. In contrast, this article did not define a “digital” tool.
Authors’ response:
Thank you for your thoughtful review of manuscript ID 2957803_revised version. We appreciate you highlighting valuable information and a key point for clarification of the manuscript.
The changes incorporated in this round of revisions (second) are highlighted in blue.
1.1) Definition of digital tools for literature extraction should be described in the Methods section.
Authors’ response:
Your point regarding the need for a clear definition of "digital tools" in the context of this review is well taken. We acknowledge that dietary assessment often involves some level of digital interaction. However, in this review, we aimed to focus on tools specifically designed for data collection and analysis within the digital environment.
Revised Definition:
To address this, we incorporated a definition of "digital tools" for dietary assessment within the Methods section. The text now reads “In this review we define digital tools as electronic platforms or applications specifically designed to collect, store, and analyse dietary intake data. These tools may include features such as self-administered questionnaires, image capture, barcode scanning, and integration with dietary databases for nutrient estimation” (page 2, lines 71-75) in the revised manuscript (Round 2).
Based on our definition of digital tool, studies solely relying on paper-based questionnaires that are later manually entered into spreadsheets or other software were not included. (page 2, lines 77-79) in the revised manuscript (Round 2).
1.2) Literatures with unclear tools should be deleted from the list. Supplementary Table 2 have 89 references (it may be in column “ID”), which is less than 94 studies. Most readers, like this reviewer, cannot read Portugal, or Spanish literatures. Clarify the digital process in each tool.
Authors’ response:
Supplementary Table Discrepancy
You're correct in pointing out the discrepancy between the number of studies reported in the text (94) and the number of references listed in Supplementary Table 2 (89). We apologise for this oversight. We carefully reviewed the table to ensure it accurately reflects the number of included studies and provide clarification for any missing references.
We critically reviewed and reorganised all of the material and divided it into three tables provided as Supplementary Material (Supplementary Tables S1, S2 and S3), namely S1 - description of the mapped studies (N=94); S2 - description of the digital tools (N=48); and S3 - studies related to the validation of the tools (N=11).
Clarifying Digital Processes
We acknowledge that the current manuscript may not fully elucidate the digital processes employed by each included tool. We have included a brief description of each tool's digital features in the results section, and a separate set of supplementary tables provides details of the results available for each of the selected studies. This will give the reader a clearer understanding of each tool's specific digital aspects.
Addressing Language Barriers
We realise that the inclusion of Portuguese and Spanish literature can create a barrier for some readers. To mitigate this problem, we have included the objectives of each study in Table S2 and tried to improve the description of the validation studies in Table S3 (Supplementary Material).
- Duplicated literature which used the same sample should deleted. Most readers, like this reviewer, cannot read Portugal, or Spanish literatures. For example, Ref. 98 used the subsample data. Ref. 68 may be thesis. Such article might be often published in the journal after that. Such cases are sometimes seen in the Reference section. Additionally, the method to delete duplication should be described in the Methods. This was not included in the reason in Figure 1.
Authors’ response:
We were careful to avoid potential duplication in cases where a thesis (e.g. Ref. 68) might subsequently be published in a journal. Whenever possible, we prioritized the articles published in such cases. We describe this approach in the Methods section. The PRISMA flow diagram shows the 81 duplicate records removed during the results extraction process (Figure 1, page 5, line 134). The presence of dissertations and theses among the mapped studies is part of the search strategy of a scoping review, whichshould include papers (primary studies) both published and unpublished (grey literature) (Levac, Colquhoun, O’Brien, 2010).
As you mentioned, the inclusion of Portuguese literature may create a barrier for some readers. While non-Portuguese or Spanish speaker researchers might find it difficult to access the original paper cited in this review, the main characteristics of included studies, their limitations and key results are presented in the manuscript. Additionally, the majority of included papers have English abstracts. Conducting a review solely with English literature would significantly limit its comprehensiveness and would hinder a complete picture of the landscape in this Brazil.
We believe that by incorporating these revisions, we can significantly enhance the clarity and comprehensiveness of our manuscript. Thank you again for your valuable feedback.
Reference:
Levac D, Colquhoun H, O’Brien KK. Scoping studies: advancing the methodology. Implementation Science. 2010; 5(69):2-9. https://doi.org/10.1186/1748-5908-5-69
- Reference numbers are inconsistent. “Several tools demonstrated satisfactory validity for assessing dietary intake, including the Online version of the Previous Day Food Questionnaire (PDFQ) for schoolchildren [63], Web-CAAFE for students [79], QUACEB for schoolchildren [96], and CUME online FFQ for adults [27]. The QSFA showed relative validity for evaluating average calcium intake [67]. The DQI-DFG tool exhibited consistent validity and reliability [50]. NutriSim was found to be user-friendly but with scattered answers [104] (Table S2).” PDFQ cited Ref63 in “Online version of the Previous Day Food Questionnaire (PDFQ) for schoolchildren [63]”. In Table S2, PDFQ indicates Ref. 39. Ref. 39 is available at:
https://bvsms.saude.gov.br/bvs/publicacoes/vigitel_brasil_2011.pdf, but the translated PDF does not include schoolchildren, but involved only adults, +18y. Other reference numbers are also incorrect.
Authors’ response:
We acknowledge the discrepancy in the references of studies reported and have made the necessary corrections. The revised text now states, "Several tools demonstrated satisfactory validity for assessing dietary intake, including the Online version of the Previous Day Food Questionnaire (PDFQ) for schoolchildren [63], Web-CAAFE for students [79], QUACEB for schoolchildren [96], and CUME online FFQ for adults [27]. The QSFA showed that the QSFA online was validated for iron and calcium [95]. The DQI-DFG tool exhibited consistent validity and reliability [50]. NutriSim was found to be user-friendly but with scattered answers [68] (Table S3)."
The FODMAP Project tool demonstrated good reproducibility for all FODMAP groups and good validity for lactose, with lower validity for other FODMAPs [115]. The CAAFE tool/ questionnaire required further validation to improve its accuracy [59] (Table S3)." (page 7, lines 191-200) in the revised manuscript (Round 2).
Minor points
- there are 45 articles in L14, but 48 articles in 158. Table S2 have 45 articles. The percentage in L177 is wrong.
Authors’ response:
We acknowledge the discrepancy in the references of studies reported and have made the necessary corrections. The revised text now states, "This review identified forty-eight digital tools in the 94 publications analyzed." (page 1, lines 14-15) in the revised manuscript (Round 2).
- L177. “11 come from validation studies.” 11 tools had a validation test, OR 11 tools have validated comparing with a standard method.
Authors’ response:
All 11 studies described in Table S3 reported the validation of digital tools using a gold standard or reference method as a comparison method.
- The column header of Table S2, “ID”, should be changed in “Reference number, if so.
Authors’ response:
We critically reviewed and reorganised all of the material and divided it into three tables provided as Supplementary Material (Supplementary Tables S1, S2 and S3), namely S1 - description of the mapped studies (N=94); S2 - description of the digital tools (N=48); and S3 - studies related to the validation of the tools (N=11).
